# Decoupling between Economic Growth and Carbon Emissions: Based on Four Major Regions in China

**DOI:** 10.3390/ijerph20021496

**Published:** 2023-01-13

**Authors:** Tao Shen, Runpu Hu, Peilin Hu, Zhang Tao

**Affiliations:** 1The Institute for Sustainable Development, Macau University of Science and Technology, Macao 999078, China; 2School of International, Zhengzhou University, Zhengzhou 450000, China; 3School of Business, Shenzhen Institute of Technology, Shenzhen 518116, China; 4School of Logistics Management and Engineering, Nanning Normal University, Nanning 530001, China

**Keywords:** carbon emissions, economic growth, impulse response function, Tapio decoupling, VAR model

## Abstract

This paper constructs a decoupling model for four major economic regions of China, based on the Tapio decoupling index method and VAR model for carbon emissions to compare and measure the impact of decoupling between carbon emissions and economic growth in China during 1997 to 2019. The results show that the degree of decoupling between economic growth and carbon emissions varies among different economic regions, and the decoupling status is better in all regions at the beginning of the 21st century. In general, the decoupling status in the eastern and western regions is more ideal than that in the central and northeastern regions. The impulse response and variance decomposition results show that renewable energy consumption could always reduce the growth of carbon emission intensity, and its effects are most significant in the western region. The above findings help to reveal the link between economic growth, renewable energy consumption and carbon emissions in China in recent years, and how to ensure a stable economic growth in China while increasing the share of clean energy consumption in each region to achieve carbon neutrality.

## 1. Introduction

Economic growth cannot be achieved without significant energy consumption [1]. However, energy consumption is the main source of carbon emissions [2]. China actively participates in global climate governance and attaches importance to the coordination between energy consumption, economic development and carbon emissions [3]. Therefore, how to decouple carbon emissions from economic growth has become one of the key issues to be solved in China’s carbon peaking and carbon neutrality development path [4]. The development of a green economy with low-carbon characteristics requires a reduction in the consumption of non-renewable energy and an increase in the consumption of renewable energy to reduce carbon emissions [5]. China’s decoupling of carbon emissions from the economy has strong geographical characteristics [6]. Non-renewable energy consumption is environmentally unfriendly but beneficial to the stability of economic performance. Conversely, renewable energy consumption is environmentally friendly but increases the instability of economic growth due to the limitations of existing energy production technologies [7,8].

Carbon emissions are closely related to the type of energy consumption. The elasticity of carbon emissions to total oil consumption is positive and significant [9,10]. In addition, the use of coking coal has the strongest effect on the growth of carbon emissions. Therefore, the reduction of coking coal is very critical to achieve carbon neutrality goals [11]. Compared with oil and coking coal, natural gas as a clean energy source would significantly reduce CO_2_ emissions [12]. Therefore, energy gases such as natural gas have become an extremely popular energy alternative to coal in China since the last decade due to its abundant reserves and significant carbon reduction advantages [13]. In contrast to non-renewable energy sources, the uses of renewable energy sources such as hydropower, wind, and solar are seen as effective ways to achieve carbon neutrality goals [14]. However, in China, the effectiveness of the interaction between carbon emissions and renewable energy consumption is weak in both the long and short term [15]. This implies that renewable energy has not been utilized significantly to reduce CO_2_ emissions in China.

Based on the Kuznets’ inverted U-curve hypothesis and the decoupling theory of carbon emissions, the decoupling between economic growth and greenhouse gas emissions gradually occurs when the economy develops to a certain level [16,17]. For China, the degree of decoupling between carbon emissions and economic growth has been increasing in super first-tier cities [18]. Western regions such as Sichuan are dominated by renewable energy consumption with little ecological damage but unstable energy supply, while central regions such as Shanxi and Henan provinces are dominated by coal consumption with serious environmental pollution but stable energy supply [19]. Therefore, based on the energy consumption structure of different regions in China, analyzing the geographical differences of decoupling economic growth and carbon emissions within China, securing and increasing the share of renewable energy consumption and reducing the consumption of non-renewable energy in the risk of stable energy supply are important ways to promote the decoupling of economic growth and carbon emissions in China.

The research contributions of this paper are as follows. Firstly, this paper provides new evidence to the existing literature on the decoupling of economic growth from carbon emissions in China’s four major economic regions, and the dynamic effects between renewable energy consumption and carbon emission intensity. Secondly, this study constructs a decoupling model based on the accurate description of carbon emission-related indicators in different regions, which could compare and measure the decoupling effects between carbon emissions and economic growth across the regions, while introducing carbon intensity as an indicator to further verify the impact of renewable energy consumption in the context of environmental protection. Moreover, the dynamic VAR model has the advantage of better reflecting the effects of renewable energy consumption and carbon intensity in different regions. Finally, this study analyzes the actual effects of renewable energy development on energy conservation and emission reduction in each region over 23 years from a spatial perspective based on the consumption structures of renewable and non-renewable energy of the four major economic regions in China. Its advantages are to measure the impacts of carbon emissions and carbon emission intensity according to the difference of energy consumption statuses in different economic regions, and to propose carbon emissions policy recommendations according to local conditions.

The conceptual framework of this paper is shown in Figure 1.

## 2. Literature Review

In the past five years, China has become the world’s top trading and industrial country. Especially since the outbreak of the COVID-19 in 2020, with the gradual recovery of China’s economy, carbon emissions and energy consumption have increased significantly. In general, according to Annual Report on China’s Energy Development (2022), coal combustion accounts for 56% of energy consumption, but major renewable energy consumptions, such as solar and wind, register at a relatively lower share. For instance, in provinces such as Sichuan and Yunnan, the energy consumptions are mainly based on hydropower, which is environmentally friendly, but the supply stability of energy is weak and vulnerable to unexpected incidents such as ecological or geographic changes. For example, power restriction policies in Sichuan and Northeast China in the summer of 2022 had a negative impact on economic growth. There is a relatively rich literature available to trace the dynamic relationship between energy consumption, economic growth and carbon emissions, which are reviewed as follows.

### 2.1. Impact of Energy Consumption on Economic Growth and Carbon Emissions

Most scholars believe that decoupling is not yet happening in a significant way in many developing countries. The level of economic development has a key offset to the decoupling process, while energy intensity plays a significant role in promoting decoupling [20]. In developed countries, for example, the decline in energy intensity (the ratio of energy consumption to GDP) drives a strong decoupling state [21]. At the national level, energy intensity and the level of economic activity are the most important factors influencing decoupling in China’s economy. The large consumption of fossil energy is an important reverse driver for decoupling of carbon emissions from the economy [22]. During 1980–2014, China has achieved a weak decoupling of economic growth from carbon emissions [23]. After 2016, China’s regional economic development and carbon emissions have shown a stable decoupling trend, and China has embarked on a path towards a low-carbon economy [24].

Moreover, economic growth has been a main driver of carbon emissions growth in recent decades, and the optimization of energy supply and consumption structures has had a significant dampening effect on carbon emissions [25]. In terms of the impact of sectors on decoupling in China, the five major industries achieved weak decoupling in 2014 [26]. Among them, the industrial sector played a leading role [27]. The industrial and transportation sectors have significant impacts on carbon emissions reduction [28], and their contributions to energy conservation and emission reduction have great potentials. Specifically, due to the implementation of national energy conservation and emission reduction policies, the industrial and transportation sectors have gradually improved to weak decoupling with low output value and high capacity in Beijing after 2008 [29]. The application of renewable energy promotes the decoupling of economic growth and carbon emissions [30].

Therefore, increasing R&D investment in renewable energy has a positive impact on decoupling economic growth from environmental pressure. In China, this impact of increasing R&D in renewable energy is most pronounced. Other factors, such as urbanization, population density and the stability of renewable energy supply, could also influence the decoupling of economic growth from carbon emissions [31]. For example, there is a large gap in the degree of implementation of environmental policies and technologies between developed and less developed provinces, so the focus of future decoupling development might be on the promotion of energy-saving technologies, industrial structure upgrading and energy structure optimization, in addition to the strict implementation of national environmental policies [32].

### 2.2. Regional Characteristics of Carbon Emissions in China

The current overall level of carbon reduction technology in China is low, providing limited contribution to decoupling economic growth from industrial energy carbon emissions [33]. China’s carbon emissions have a strong cross-regional convergence and spatial characteristics [26]. Thus, there are stepwise differences in carbon emissions among the eight economic regions in China. The share of carbon emissions in the overall carbon emissions variance are different in different regions, whether within or without the regions [34]. With the restructuring of energy consumption, inter-regional carbon emissions differences have been increasing [35].

In the view of some scholars, economic growth is the main factor for the increase of carbon emissions [36]. There is a strong divergence relationship between GDP and carbon intensity in developed regions of China [37], while the decoupling is not obvious in other regions [38]. For example, the decoupling between GDP and carbon emissions is strong in the Yangtze River Delta and Pearl River Delta [39], but provinces such as Henan and Shanxi still show a strong positive correlation between the degree of economic development and carbon emissions. In energy-rich regions, lower energy consumption intensity would significantly reduce regional carbon emissions [40]. With the exception of Hainan, Guangxi, Ningxia and Xinjiang, the adjustment of energy structure in most regions has little impact on carbon emissions.

From the perspective of carbon emissions reduction, the uncoordinated regional economic development in turn affects the interregional differences in national environmental policies in China [41]. At the same time, it also changed the pattern of carbon emissions in China: the central economy region relying on non-renewable energy has the most prominent carbon emissions, and carbon flows between energy-rich regions in the west and economically developed regions in the east are more active [42]. The impact of increased carbon emissions is greater in regions where the energy consumption structure is dominated by coal and thermal power. This kind of impact is more pronounced in central China, followed by eastern and western regions [43]. In western and central Chinese cities, economic growth and private vehicle fuel consumption are the main causes of carbon emissions, and the higher-end industrial structure of the Beijing-Tianjin-Hebei region has a positive effect on CO_2_ levels [44]. The Kuznets curve does not hold in the central and western regions and has a weak hold in the eastern region [45].

### 2.3. Pathways of Energy Market Impact on Carbon Emissions

First of all, energy markets affect energy consumption demand through energy prices, which in turn affects carbon markets and influences carbon emissions. There is a bidirectional causal relationship between renewable and non-renewable energy consumption and economic growth [46,47,48]. Due to the improvement of market regulation and the related determination to combat climate change, China’s carbon market has strengthened its linkage with major energy markets [43]. Therefore, the price fluctuations on international energy markets have a significant impact on China’s carbon emissions. Based on the theory of information spillover and chain effects of systemic risk, the systemic risk generated by the world energy market would affect the structure of renewable and non-renewable energy consumption in China, thus causing fluctuations in domestic carbon emissions. For example, the war between Russia and Ukraine caused all the original gains in WTI crude oil futures to fall to zero, and Brent crude oil contract prices and natural gas prices have experienced several rounds of spikes, with the market showing amazing volatility. The high price of oil would not only reduce per capita oil consumption, but also force China to upgrade its industries to make enterprises cleaner and consume less, promoting China’s carbon peaking and carbon neutrality goals.

In addition, the volatility of the oil market would increase the risk of the natural gas market [49], and the international oil price would affect the domestic utilization of natural gas in China, thus infecting the carbon market with certain risks as carbon emissions tend to be unstable. The risky volatility of the crude oil market under the influence of extreme events could affect the utilization and consumption of renewable energy [50]. This would have a double negative impact on China’s carbon market. The risk transmission between carbon emissions and the electricity market is rooted in the oil market, where uncertainty in the oil market affects the price of electricity to a certain extent [51]. The high or low price of electricity affects the use of energy types for power generation, which could increase or decrease carbon emissions. Among all energy markets in China, the coking coal market has the strongest impact on carbon emissions. The increase in coal market risk and the decrease in coal utilization would have a positive spillover effect on the carbon market and reduce carbon emissions. The stable development of the coking coal market has a very critical role in carbon neutrality goals [11]. 

In terms of contents, unlike most previous articles, according to the division of China’s four economic regions by the National Bureau of Statistics, this paper makes use of the time series data of different regions to conduct research from a macro perspective, so as to better put forward policy suggestions by comparing the situation of different economic regions. This paper does not study the dynamic relationship between energy consumption, carbon emissions and economic growth, as most other literature does by considering China as a whole. Four economic regions are divided in detail in this paper. Based on the decoupling relationship between renewable energy consumption, carbon emissions and economic growth in each economic region, this paper studies regional green development and puts forward suggestions. This makes up for the lack of empirical research on renewable energy consumption in sub-regions. Moreover, there are many papers on the relationship between renewable energy consumption, carbon emissions and economic growth, but they ignore the relationship between renewable energy consumption and carbon emission intensity. This paper makes up for this gap. Additionally, different from previous studies, this paper constructs a dynamic VAR model to study the relationship between renewable energy consumption and carbon emission intensity.

## 3. Data and Models

### 3.1. Data Sources

In this paper, based on National Bureau of Statistics, according to the new situation of China’s accelerated economic and social development, China is divided into a northeastern region, eastern region, central region and western region. Each region is composed of different provinces, which have certain differences in geographical location, energy structure and economic development. There are no independent judicial and legislative powers in different provinces. Moreover, the administrative power needs to be influenced by the superior, so the country should formulate different development strategies according to the local conditions of different regions. 

Time series data from 1997–2019 are selected for each economic region, mainly from the China Energy Statistical Yearbook (CESY) and the Carbon Emission Accounts & Datasets (CEADs). The China Energy Statistical Yearbook is a data book that comprehensively reflects the balance of China’s energy construction, production, consumption and supply and demand, and is mainly edited by the Department of Energy Statistics of the National Bureau of Statistics. The carbon emissions of the four major regions in China are calculated based on China’s emission apparent accounting method by referring to the research of Shan et al. [52]. Data from Tibet Autonomous Region, Hong Kong, Macao and Taiwan are not included due to data limitation. The EIC represents the carbon emission intensity, which refers to the carbon dioxide emission per unit of GDP, and the carbon emissions are divided by the actual GDP of each year to get the carbon emission intensity of the corresponding year. In addition, the data and variable descriptions are shown in Table 1.

### 3.2. Model Setting

Due to the great differences in economic development, energy policies and carbon emissions in different regions of China, this paper divides China into four economic regions, using the Tapio decoupling model to study the decoupling of economic growth and carbon emissions in different economic regions, and makes a comparative analysis in horizontal regions and longitudinal time. At the same time, many studies have proved that the development of renewable energy is beneficial to environmental protection, carbon reduction and sustainable environmental development. However, due to the constraints of environmental resources, geographical location and other factors in different economic regions, the consumption of renewable energy in different regions varies greatly. Therefore, this paper introduces the variable renewable energy consumption in four economic regions. Since decoupling means reducing environmental pressure during economic growth, in order to better measure this index, this paper introduces the variable ‘carbon emission intensity’, namely the carbon dioxide emission per unit of GDP. Then, this paper uses the impulse response function based on VAR model to study the dynamic effect between renewable energy consumption and carbon emission intensity by region.

#### 3.2.1. Decoupling Model

The term “decoupling” originates from the field of physics and is mainly used to describe the coupling state between variables. Decoupling analysis has become a hot topic of research in the field of resources and environment; “decoupling” means to reduce the consumption of material resources or environmental pressure while maintaining economic growth. This method has been widely used in different studies. For example, Zhang et al. [53] used the Tapio decoupling index method to analyze the decoupling relationship between China’s economic development and energy consumption. Gao et al. [54] used the Tapio decoupling index method to analyze the relationship between transportation energy consumption and economic development. Currently, three methods are widely used to measuring the state of decoupling, which are the decoupling factor method proposed by OECD, the decoupling elastic coefficient method proposed by Tapio and the decoupling evaluation method based on IPAT equation. The Tapio decoupling model mainly measures the degree of decoupling among variables by calculating the elastic coefficient. The Tapio decoupling model can not only analyze the influence of various factors on the decoupling index by constructing causal chain, but also effectively overcome the influence of the change of metrology dimension. Therefore, this paper adopts the Tapio decoupling analysis model. The decoupling relationship between economic growth and carbon emissions in different regions is defined as follows:(1)$$T\left(C,G\right)=\frac{{(C}_{t+1}-C_{t})/C_{t}}{({GDP}_{t+1}-{GDP}_{t})/{GDP}_{t}}$$
where *T* is the decoupling coefficient, $C_{t+1}$ and $C_{t}$ denote the total carbon emissions of different regions in periods *t* + 1 and *t* and ${GDP}_{t+1}$ and ${GDP}_{t}$ denote the GDP of different regions in periods *t* + 1 and *t*, respectively.

According to the different decoupling coefficients, decoupling is generally divided into three major decoupling types, and eight decoupling statuses (shown in Table 2). The most desirable state is the strong decoupling state, which indicates positive economic growth and negative carbon emission growth, i.e., economic growth is accompanied by a decrease in environmental pressure. The most undesirable state is strong negative decoupling, which means negative economic growth with increasing environmental pressure, which is the opposite of strong decoupling. Different regions should aim at achieving strong decoupling as an ultimate goal.

#### 3.2.2. VAR Model

VAR is commonly used to describe the effects of stochastic perturbations on a system of variables. The modeling idea of VAR model is that each endogenous variable is constructed as a function of the lagged terms of all endogenous variables in the system. Then, it can be extended to a vector autoregressive model consisting of multivariate time series variables to estimate the dynamic relationships among all endogenous variables. This paper analyzes and predicts the degree of mutual shock impact between renewable energy consumption and carbon emission intensity in each of the four major economic regions by building a dynamic VAR model; the basic form is as follows:(2)$$Y_{t}=A_{0}+A_{1}Y_{t-1}+\ldots +A_{p}Y_{t-p}+\delta_{t}$$

In order to eliminate the heteroskedasticity problem of the time series, the variables RE and EIC are logarithmically processed, which can reflect the variation of the elasticity coefficients among the variables without changing the cointegration relationship between the original time series, and the following results are obtained:$$Y_{t}=\{LNRE,LNEIC\}$$
where $Y_{t}$ denotes an m-dimensional non-smooth $I\left(1\right)$ sequence and $Y_{t}$ is a two-dimensional endogenous variable in this paper. $A_{0}=$0,$A_{1}$,⋯,$A_{p}$ are nth-order coefficient matrix. $\delta_{t}$ is a random perturbation term, which is a white noise vector.

### 3.3. Estimation Method

#### 3.3.1. Impulse Response Analysis

Before model construction, unit root test and cointegration test need to be performed on the time series data. This paper adopts the Augmented Dickey–Fuller method to test the smoothness of the time series of carbon emission intensity and renewable energy consumption in China from 1997 to 2019 by sub-region. It is also necessary to conduct cointegration test on the time series data, and this paper adopts the AEG two-step method to test whether there is a cointegration relationship between carbon emission intensity and renewable energy consumption by sub-region. The specific steps are as follows: in the first step, ordinary least squares are used to do linear regression of a variable on other variables, and to obtain the residual series. In the second step, a smoothness test is conducted on the residual series obtained in the first step. If the residual series is not smooth, there is no cointegration relationship between multiple variables. If the residual series is smooth, there is a cointegration relationship between multiple variables.

Impulse response function serves to study the dynamic impact process of other variables in the model in the current and later periods when the VAR model is hit by the standard deviation of one random error term, which can portray the dynamic interaction between variables and their effects in a more intuitive way. This method has also been widely used in this topic. For example, Soytas et al. [55] used generalized impulse response functions to measure the relationship among energy consumption, income and carbon emissions in the United States. This paper uses impulse response function and variance decomposition based on VAR model to study the dynamic effect between renewable energy consumption and carbon emission intensity in major economic regions of China. According to $Y_{t}=A_{0}+A_{1}Y_{t-1}+\ldots +A_{p}Y_{t-p}+\delta_{t}$, the following impulse response function can be obtained:(3)$$\tau_{n}\left(\gamma_{ik,n}\right)=\sum_{j=1}^{n}\tau_{n-j}A_{j}$$

#### 3.3.2. Variance Decomposition

Variance decomposition is a method for studying the contribution of shocks to predict the variance of each variable to all endogenous variables in a VAR model. It can explain the extent to which changes in variables are caused by their own factors and the extent to which they are caused by shocks to other factors in the system. The variance decomposition describes information about the relative importance of each random disturbance that affects the variables in the VAR model. The impulse response function simply shows how each variable in the model responds to shocks over time, so if the importance of the influence relationship between variables is analyzed, then variance decomposition is required. That is, the variance decomposition can analyze how much the first variable itself and the other variables contribute to the change in the first variable in the future m periods respectively, in the given future m periods. The variance decomposition can be divided into the traditional orthogonal variance decomposition and the generalized prediction error variance decomposition, and the former is more commonly used at present. In this paper, the orthogonal variance decomposition is used to study the degree and mechanism of the mutual influence between renewable energy consumption and carbon emission intensity by sub-region.

## 4. Results and Discussion

The descriptive statistical analysis of different regional variables is carried out, and the trend chart of the statistical results is shown in Figure 2. It can be seen that for carbon emission and GDP, all the four economic regions are on the rise. Compared with the overall trend, the economic growth of the eastern region is the fastest and the corresponding carbon emission is also higher, while the economic growth of the northeast region is slower than the other three regions. As for the consumption of renewable energy, all regions show an upward trend. With the change of time, the western region is better than the eastern, central and northeast regions, which is related to the rich hydropower and wind power resources in the western regions, such as Sichuan and other provinces, and the high proportion of renewable energy consumption, reflecting the differences in the energy structure among different regions. As for the carbon emission intensity, the four regions show a downward trend and are gradually approaching each other, which reflects that China’s carbon emissions per unit of economic growth are becoming less and less, and the differences between regions are also gradually decreasing, which is related to the technological progress brought about by economic growth.

### 4.1. Results of the Decoupling Model

The trend diagram of economic growth rate and carbon emissions rate over time for the four major economic regions in China is shown in Figure 3, and the decoupling by regions is shown specifically in Table 3. As can be seen from Figure 3, carbon emissions of the northeast region experienced negative growth in 1998, 2001, 2007, 2013–2015 and 2017, while the economic growth rate was positive from 1998 to 2015, but the GDP experienced a negative growth from 2016 to 2019. In general, changes in economic growth and carbon emissions are not yet stable enough and are highly fluctuating. As for the eastern region, which has more developed provinces, GDP has been maintaining positive, with rapid growth from 2004 to 2011, corresponding to the same period of carbon emissions, which are also in a high growth state. Both of growth rates have slowed down in recent years, and the growth rate of carbon emissions is always lower than the economic growth rate. As the central region with a high proportion of traditional energy consumption, the GDP has always maintained positive growth. For the western region, the GDP growth is always positive, but the overall growth rate is slower than that of the central region, and the growth rate of carbon emissions is slower than that of the economy, except for some years when it is faster than the GDP growth rate.

The decoupling results are shown in Table 3, where WD stands for weak decoupling, SD stands for strong decoupling, END stands for expansive negative decoupling, SND stands for strong negative decoupling and EC stands for expansive coupling. As can be seen from Table 3, for the northeast region, the decoupling state is not stable enough. The state is always best in the strong decoupling state from 2013 to 2015, and in the strong negative decoupling state in 2016 and 2019. It shows that the GDP has negative growth, but the growth rate of carbon emissions is still positive. For the eastern region, decoupling is dominated by weak decoupling, but strong decoupling was achieved in 1998, 2013 and 2017, respectively, and the decoupling status in recent years has been better than that in the early 20th century. This reflects a reduction in the growth rate of carbon emissions while maintaining economic growth. For the central region, five strong decoupling were achieved from 1998 to now, three of them were from 1998 to 2000, when the growth rate of the central region was relatively slow and the impact of economic growth on the environment was not obvious enough. The last two moments of strong decoupling arose from the negative growth of carbon emissions achieved under the faster economic growth, reflecting the positive effect of economic growth on the environment. There was mainly weak decoupling in the western region, and a total of three strong decoupling moments have been achieved, though only one strong decoupling moment in the last five years. The overall growth rate of carbon emissions decreased, but the positive effect of economic development on the environment is still not obvious.

### 4.2. Unit Root Test and Cointegration Analysis Results

The unit root test for each variable shows that the first-order difference series of the original time series is smooth, so it can be concluded that the time series of LNRE and LNEIC of the four major economic regions is a first-order single integer. The results of the cointegration analysis are shown in Table 4, which shows that the regression residual series of LNRE and LNEIC in northeast region is not smooth, so there is no cointegration relationship between renewable energy consumption and carbon emission intensity in the northeast region. The regression residual series of LNRE and LNEIC in the eastern, central and western region are smooth, so there is no cointegration relationship between renewable energy consumption and carbon emission intensity in these three major economic regions. Therefore, there is a cointegration relationship between renewable energy consumption and carbon emission intensity in these three major economic regions.

### 4.3. Estimation Results of VAR Model

Since no cointegration relationship exists for the northeastern time series data, it is not suitable for further modeling analysis. There is a co-integration relationship between renewable energy consumption and carbon emission intensity in the eastern, central and western regions. The VAR models are estimated by taking the third-order, second-order and second-order lags for the eastern, central and western regions, respectively.

From the eastern region, the renewable energy consumption with the second- and third-order lags has a significantly negative effect on the carbon emission intensity, as shown in Table 5. The carbon emission intensity with the first-, second- and third-order lags also has a significantly negative effect on the renewable energy consumption. From the central region, the second-order lagged renewable energy consumption has a significantly negative effect on carbon emission intensity. In the western region, both the first- and second-order lagged renewable energy consumption have a significantly negative effect on the carbon emission intensity.

Further AR view tests were performed on the models of each of the three regions. The results show that the inverse of the mode of all characteristic roots is less than 1. They are all within the unit circle curve, which indicates that the model is stable and statistically sound, and could ensure the validity of impulse response analysis, variance decomposition and the reasonableness of data interpretation.

### 4.4. Granger Causality Test Results

In order to determine the causal relationship between renewable energy and carbon emission intensity in different regions, a Granger causality test is conducted separately. From the results of the eastern region, renewable energy consumption and carbon emission intensity show a two-way causal relationship. From the results of the central region, renewable energy consumption is the Granger cause of carbon emission intensity, but carbon emission intensity is not the Granger cause of renewable energy consumption. It shows that there is a one-way causal relationship. From the results of the western region, renewable energy consumption is the Granger cause of carbon emission intensity, but carbon emission intensity is not the Granger cause of renewable energy consumption. It indicates that there is a one-way causal relationship. 

### 4.5. Results of Impulse Response Analysis

The results of impulse response analysis are shown in Figure 4. For the eastern region, LNEIC has a strong response to the shock of its own one standard new spread immediately, with an impact rate of 0.031 in period 1, reaching a maximum value of 0.036 in the subsequent period 2, then decreasing from period 3 to period 5, reaching a minimum value of 0.013 in period 5, and then leveling off. LNEIC does not respond to the shock of LNRE one standard new spread in the current period, and then does not respond until period 3. LNRE has a strong response to the shock of its own one standard new spread immediately, reaching 0.12, and then fluctuates under 0.12 to reach a minimum value of 0.026 in period 4, and then tends to be stable, and the shock of LNRE to LNEIC one standard new spread reaches a maximum value of 0.019 in period 6, and then tends to be stable. Shock reaches a maximum value of 0.015 in the current period, then drops to a negative value in period 2, reaches a minimum value of 0.024 in period 4 and then plateaus.

For the central region, LNEIC has a strong response to the impact of its own one standard new spread immediately, with an impact rate of 0.081 in period 1, then gradually falling to a low of −0.005 in period 6 before returning to a positive value of 0.006 in period 7, and then leveling off, which shows that LNEIC adjusts more slowly to the impact of its own one standard new spread. The shock of LNEIC to LNRE’s one standard new spread does not respond in the current period, rises to a high of 0.024 in period 2, then drops to a negative value in period 3, reaches a low of −0.056 in period 6, and then stabilizes. The shock of LNRE to its own one standard new spread reacts strongly in the current period, reaching 0.156, then slowly fluctuates down and stabilizes after period 6. The shock of LNRE to one standard new spread of LNEIC reaches a minimum of −0.029 in the current period and then fluctuates and adjusts to −0.028 in period 5 before leveling off, which is not strong enough to explain itself.

For the western region, LNEIC has a strong response to the shock of its own one standard new spread immediately, and starts to decline after reaching the highest point of 0.037 in period 2, reaching the lowest point of 0.015 in period 6, and then reflecting a smooth convergence to 0. LNEIC does not respond to the shock of LNRE one standard new spread in the current period, and then starts to decline, reflecting a negative shock effect, reaching the lowest point of −0.038 in period 5 after reaching a low of −0.038, after which it leveled off. LNRE’s shock to its own standard new spread immediately had a strong response, falling to 0.079 in period 2 and then rising to a high of 0.093 in period 5, then leveling off. LNRE’s shock to LNEIC’s standard new spread reached −0.01 in the current period, then rose to a positive effect, reaching a high of 0.017 and then leveled off, with the overall shock effect close to 0 and not obvious enough.

### 4.6. Results of the Variance Decomposition

The results of the variance decomposition are shown in Figure 5. From the variance decomposition of LNRE in the eastern region, the contribution rate of LNRE itself slowly decreases from 98% at the beginning to 85%, while the contribution rate of LNEIC slowly increases from 2% at the beginning to 15%. It indicates that as the carbon emission intensity decreases, the endogenous promotion effect of renewable energy consumption gradually decreases, and the influence of carbon emission intensity on its consumption gradually increases. From the variance decomposition of LNEIC in the eastern region, the contribution rate of LNEIC itself tends to 100% in the first three periods, and the contribution rate of LNRE to it is basically equal to 0. From the fourth period onwards, the contribution rate of LNEIC itself gradually decreases, and the contribution rate of LBRE gradually increases to 33% in the tenth period, which indicates that as time goes by, the promotion effect of vigorous development of renewable energy on the reduction of carbon emission intensity becomes more obvious.

From the variance decomposition of LNRE in the central region, the contribution rate of LNEIC to LNRE has been maintained at a low level, and the explanation effect is not obvious. From the variance decomposition of LNEIC in the central region, the contribution rate of LNRE to LNEIC has been maintained at a low level in the first three periods, and started to climb rapidly in the fourth period, with an obvious upward trend. This indicates that in the process of carbon emission intensity reduction, the increase of renewable energy consumption has a positive influence on the process of decreasing carbon emission intensity.

From the variance decomposition of LNRE in the western region, the contribution rate of LNEIC to LNRE has been maintained at a low level, and the explanation effect is not obvious. From the variance decomposition of LNEIC in the western region, the contribution rate of LNRE to LNEIC has been maintained at a low level in the first two periods, and started to climb rapidly in the third period, with an obvious rising trend. It shows that in the process of decreasing carbon emission intensity, the increase of renewable energy consumption has a positive effect. 

As a whole, the energy market is closely related to carbon emissions, with the oil, coal and clean energy markets being closely linked. Thus, price fluctuations in any one of them may have spillover effects on the supply and demand of clean energy, and thus affect carbon emissions. Therefore, preventing potential risks in energy markets is also of great significance for carbon reduction.

### 4.7. Discussion

According to the decoupling analysis results, the decoupling index of carbon emissions and economic growth in China’s four major economic regions from 1998 to 2001 was the most ideal, which is consistent with the relevant research results of Zhang and Da (“[53]”). The formulation and persistent implementation of corresponding carbon reduction policies indeed have a positive impact on the decoupling of GDP and carbon emissions. In terms of the overall situation of China from 2002 to 2008, carbon emissions and economic growth in major economic regions of China mostly showed weak decoupling and expansive negative decoupling, which is consistent with the relevant studies of Wang and Jiang [56], and a small part of the time showed expansive coupling in individual regions. However, most studies have not made a further comparative analysis by region. From the perspective of the comparison by region, the decoupling results in the central region were even less ideal in this period. At this stage, a large number of key projects were concentrated in China and the rapid economic development also brought about the improvement of energy consumption intensity, which may restrain the decoupling of the GDP and carbon emissions. From 2009 to 2019, the central, eastern and western regions were more significantly decoupled in the later period than in the earlier period. Weak decoupling was dominant, and strong decoupling was ideal in some years, indicating that the beneficial impact of technology on the environment brought about by economic growth was gradually emerging. The overall decoupling situation in the eastern region is more stable than that in other regions, which may be related to the more developed provinces and the more advanced and rational energy structure in the east region. However, at the same time, the decoupling situation in the northeast was not satisfactory in this period, and even in 2016, there was still a strong negative decoupling status, which was related to the fact that the economic growth in the northeast has slowed down in recent years, but the results of energy conservation and emission reduction strategies were in conclusion. In the future, more attention should be paid to optimizing the energy structure in the northeastern region and minimizing carbon emissions on the premise of ensuring economic growth.

According to the results of the co-integration test, the impulse response analysis and variance decomposition, there is a co-integration relationship between the renewable energy consumption and carbon emission intensity in the central, eastern and western regions. However, there is no long-term co-integration relationship between the two variables in the northeast region. This may be related to the low proportion of renewable energy consumption in the northeast, and its effect on reducing carbon emission intensity is not obvious. In the future, more attention should be paid to the energy structure of this region. The results of the impulse response analysis and variance decomposition show that renewable energy should be developed in the future, which will play a positive role in the optimization of energy structure in different regions and the reduction of carbon emission intensity.

## 5. Conclusions and Policy Suggestions

### 5.1. Conclusions

This paper draws the following conclusions through the empirical analysis and testing. Firstly, the decoupling status was generally better from 1998 to 2000, from the perspective of the four economic regions as a whole. This was due to the introduction of the “fifteen small enterprises” and “new five small enterprises” regulations during that period, which promoted the decoupling of carbon emissions. With the rapid economic growth, China’s overall decoupling status has changed, strong decoupling has also emerged in various economic regions with weak decoupling prevailing in recent years. Secondly, the decoupling status of the northeastern region is less than ideal and even negative decoupling has occurred in recent years, from the comparison of the four major economic regions. The eastern and western regions have a high proportion of renewable energy consumption, and the decoupling situation has been more ideal and stable over time. The economic growth of the central region is fast, but carbon emissions tend to grow faster than economic growth, the overall decoupling situation is not ideal. Thirdly, the impulse response analysis shows that the continuous development of renewable energy for the optimization of energy structure in different regions always maintain a long-term positive effect. Carbon emission intensity by renewable energy consumption would fluctuate in a short period of time, but maintain a stable and highly negative effect in the long term. Finally, based on the variance decomposition analysis, this paper concludes that the contribution of renewable energy consumption to carbon emission intensity in the three major economic regions has been increasing over time, with the fastest change in the western region. The explanation of carbon emission intensity to renewable energy also increases slowly in the eastern region, and it is not obvious enough in the central and western regions.

### 5.2. Policy Suggestions

First of all, China should actively respond to climate change, promote carbon emission reduction and achieve sustainable and healthy economic development according to the differences in factor endowments of different economic sub-regions. The eastern region could make use of its abundant capital to carry out a large amount of clean energy technology research and development, improve green transportation in smart cities and establish industrial energy utilization efficiency. The central region could rely on clean energy cooperation and development strategy of the Belt and Road to improve policy subsidies and tax rebates for renewable energy and natural gas and focus on implementing the policy of cutting excessive capacity. To achieve industrial upgrading of high energy-consuming enterprises, the central region may should realize green and low-carbon development. The western region is rich in clean energy resources, such as solar energy, wind energy and hydropower. Therefore, it needs to attract social investment with diversified financing such as PPP mode, and promote energy management with EMS mode to make the development and production of renewable energy in the west more intensive, professional and large-scale, while curbing production costs. Fossil energy consumption in the northeast is an important support for economic development. As an important hub for Sino-Russian natural gas cooperation projects, the northeast could take advantage of its unique location to gradually increase the proportion of natural gas consumption and effectively reduce carbon emissions.

In addition, policymakers should improve China’s clean energy market risk monitoring and early warning mechanism to ensure stable supply of renewable energy. Affected by the COVID-19 epidemic and external extreme events such as the Russia-Ukraine war and natural disasters, the clean energy market is highly vulnerable to price fluctuations and jump risks. On the one hand, it is necessary to form effective price warnings, energy futures and establish a good national energy reserve to prevent major clean energy risks. On the other hand, social capital could be guided to increase capital investment in renewable energy technologies to make the supply of renewable energy more stable and secure, and to reduce power restrictions caused by technical reasons, natural disasters and so on.

The regional carbon emission differences are considered comprehensively to coordinate effectively clean energy consumption. A unified national clean energy market should be established. In clean energy-rich regions, it is necessary to fully consider the impact of adverse factors on carbon emissions. For example, the eastern region should optimize energy management to reduce the systemic risk in the process of west–east power transmission. Then, the construction of a national carbon market should be strengthened. The regulation and initiative of the market should be improved by accelerating the legislative process, improving the regulatory mechanism, introducing more market players and trading varieties and further improving the carbon accounting system.

Moreover, the leading role of the government in energy conservation and emission reduction should be played. Carbon peaks and carbon neutrality could not be achieved without the government’s efforts of industrial planning. The government should accelerate the transformation of green development, implement comprehensive conservation strategies, develop green and low-carbon industries, advocate green consumption and promote the formation of green and low-carbon modes of production and lifestyle. Carbon peaking and carbon neutrality have been actively and steadily promoted. Based on China’s advantages in energy resources, the government should further promote the energy revolution, strengthen the clean and efficient use of coal, accelerate the planning and construction of new energy systems, actively participate in global climate change governance and achieve the dual carbon goal. 

### 5.3. Study Limitations and Recommendations

There are many papers in this field, and it is difficult to find new research perspectives theoretically. Data applied in this paper are secondary data. The situation in the four economic regions is bound to be limited by geographical location, resource richness and other factors. This paper only considers a macro perspective, without considering the micro complexity, and a follow-up study should pay more attention to the issues between different regions.

### 5.4. Suggestions for Future Research

There is much more to study in the area of carbon emissions, economic growth and renewable energy consumption, which can theoretically be studied at both micro and macro levels, e.g., the impact of a specific renewable energy consumption on the decoupling of carbon emissions and economic growth can be studied and the mechanism and transmission path of systemic risk among the three can also be deeply studied. Regarding methodology, an empirical model can be introduced across disciplines to further measure the degree of interaction between them.

## Figures and Tables

**Figure 1 ijerph-20-01496-f001:**
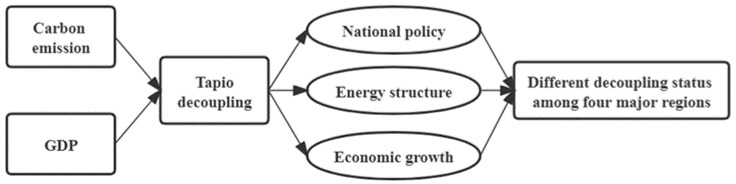
Conceptual framework.

**Figure 2 ijerph-20-01496-f002:**
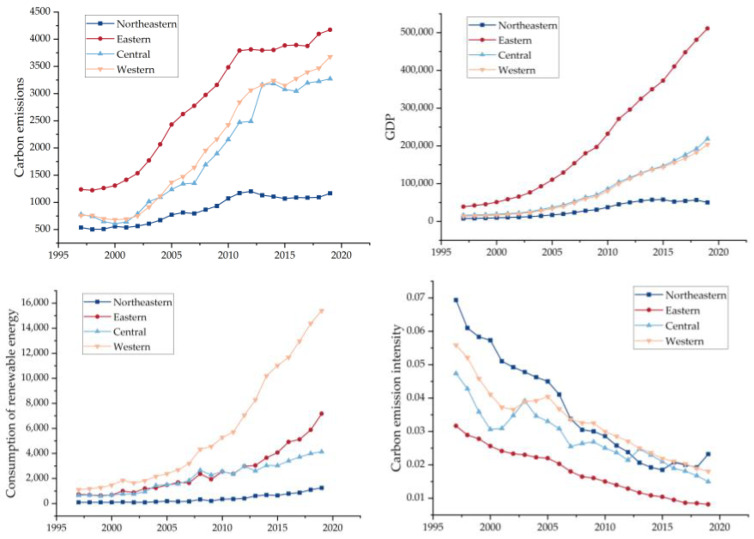
Descriptive statistical trend of variables by region.

**Figure 3 ijerph-20-01496-f003:**
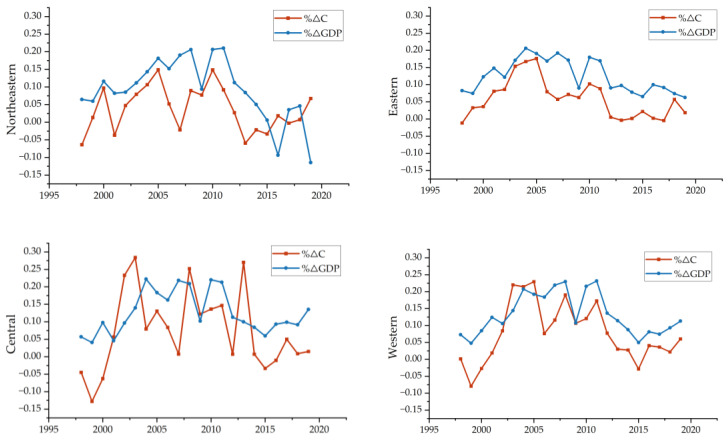
Trend of the GDP growth rate and carbon emission growth rate in four economic regions.

**Figure 4 ijerph-20-01496-f004:**
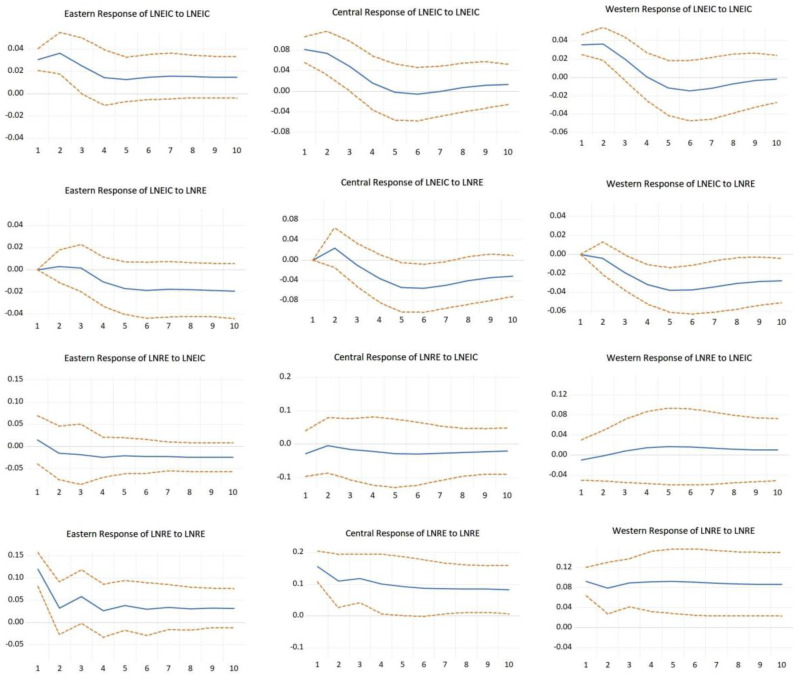
Impulse response results for the three economic regions.

**Figure 5 ijerph-20-01496-f005:**
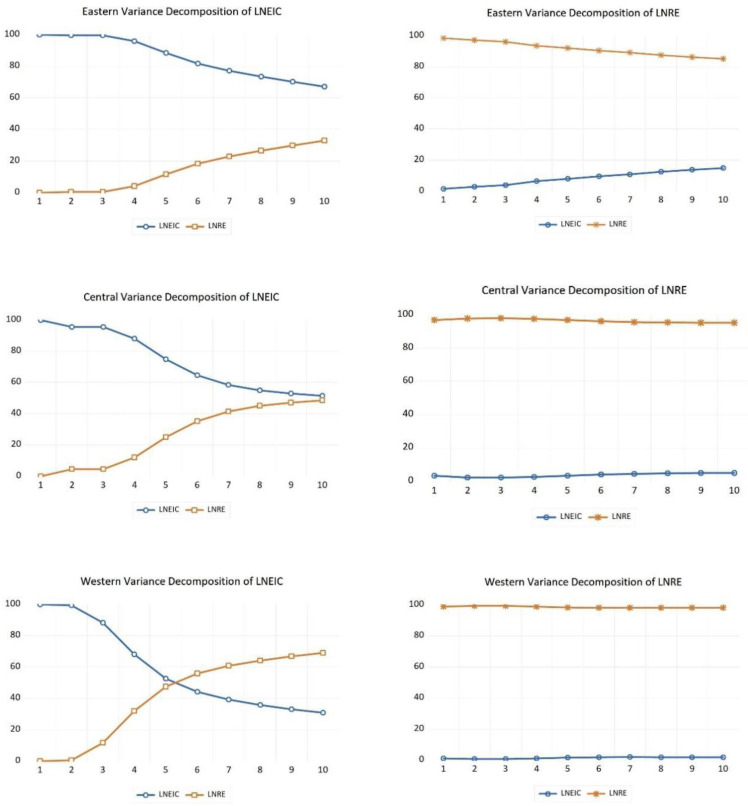
Variance decomposition results of the three economic regions.

**Table 1 ijerph-20-01496-t001:** Description of data and variables.

Variable	Unit	Source
C	Carbon emissions	CEADS
GDP	Economic development	CESY
RE	Consumption of renewable energy	CESY
EIC	Carbon emission intensity	C/GDP

**Table 2 ijerph-20-01496-t002:** Tapio decoupling evaluation criteria.

Decoupling Type	C Growth Rate	GDP Growth Rate	T-Value	Decoupling Status
Decoupling	>0	>0	0 < T < 0.8	Weak decoupling
	<0	>0	<0	Strong decoupling
	<0	<0	>1.2	Recessive decoupling
Negative decoupling	>0	>0	>1.2	Expansive negative decoupling
	>0	<0	<0	Strong negative decoupling
	<0	<0	0 < T < 0.8	Weak negative decoupling
Coupling	>0	>0	0.8 < T < 1.2	Expansive coupling
	<0	<0	<0	Recessive coupling

Note: Tapio decoupling are divided three types by T-Value.

**Table 3 ijerph-20-01496-t003:** Decoupling of carbon emissions and economic growth in four economic regions.

Year	T_(C,G)_				Decoupling Status			
	Northeastern	Eastern	Central	Western	Northeastern	Eastern	Central	Western
1998	−0.985	−0.142	−0.787	0.017	SD	SD	SD	WD
1999	0.222	0.438	−3.132	−1.668	WD	WD	SD	SD
2000	0.836	0.293	−0.642	−0.322	EC	WD	SD	SD
2001	−0.446	0.545	1.230	0.151	SD	WD	END	WD
2002	0.554	0.706	2.414	0.798	WD	WD	END	WD
2003	0.709	0.897	2.029	1.530	WD	EC	END	END
2004	0.742	0.812	0.358	1.034	WD	EC	WD	EC
2005	0.821	0.923	0.711	1.194	EC	EC	WD	EC
2006	0.341	0.469	0.517	0.414	WD	WD	WD	WD
2007	−0.115	0.299	0.034	0.529	SD	WD	WD	WD
2008	0.435	0.418	1.205	0.829	WD	WD	END	EC
2009	0.823	0.695	1.193	0.988	EC	WD	EC	EC
2010	0.719	0.569	0.620	0.558	WD	WD	WD	WD
2011	0.437	0.522	0.688	0.745	WD	WD	WD	WD
2012	0.240	0.058	0.064	0.567	WD	WD	WD	WD
2013	−0.706	−0.038	2.700	0.262	SD	SD	END	WD
2014	−0.430	0.020	0.084	0.313	SD	WD	WD	WD
2015	−5.560	0.333	−0.560	−0.574	SD	WD	SD	SD
2016	−0.195	0.021	−0.111	0.498	SND	WD	SD	WD
2017	−0.078	−0.049	0.505	0.485	SD	SD	WD	WD
2018	0.157	0.773	0.094	0.234	WD	WD	WD	WD
2019	−0.588	0.292	0.110	0.532	SND	WD	WD	WD

**Table 4 ijerph-20-01496-t004:** ADF test results of the residual series of the four economic regions.

Economic Regions	T-Value	Probability
Northeastern	−0.901	0.7877
Eastern	−3.042	0.0328
Central	−2.983	0.0365
Western	−3.601	0.0057

**Table 5 ijerph-20-01496-t005:** Estimation results of VAR models for the three economic regions.

Economic Regions	Variables	First Order Lag		Second Order Lag		Third Order Lag	
		LNRE	LNEIC	LNRE	LNEIC	LNRE	LNEIC
Eastern	LNRE	0.5092 ***	−0.8879 ***	0.5857 ***	−0.7225 **	0.4996 ***	−0.8401 ***
	LNEIC	−0.0474	0.9241 ***	−0.1179 *	−0.8207 ***	−0.1841 **	0.7396 ***
Central	LNRE	0.9274 ***	−0.0645	0.9007 ***	0.0359		
	LNEIC	−0.0924	0.7632 ***	−0.2761 ***	0.2983		
Western	LNRE	1.0584 ***	0.1912	1.1781 ***	0.5747		
	LNEIC	0.1356 ***	0.5964 ***	0.3131 ***	0.0673		

Note: ***, **, * are 1%, 5%, 10% significant levels, respectively.

## Data Availability

Data available on request from the authors.

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
