# Peer review of "Decoupling between Economic Growth and Carbon Emissions: Based on Four Major Regions in China"

_ijerph, 2023, doi:10.3390/ijerph20021496_

Round 1

Reviewer 1 Report

The paper presents interesting findings that focuses on economic growth and carbon emissions. However, the extent to which the study is different from the following studies is not clear:

·      Nonrenewable energy, renewable energy, carbon dioxide emissions and economic growth in China from 1952 to 2012

·      Economic development and carbon dioxide emissions in China: Provincial panel data analysis

·      Energy investment, economic growth and carbon emissions in China—Empirical analysis based on spatial Durbin model

The paper presents no major gap in the literature that warrant the study. 

The study does not present any defined research questions and the methodology that tie all the variables.

The paper presents no conceptual framework as the study relies on secondary data.

The literature review section did not capture studies in the area particularly from China. For example:

·      Factors influencing the progress in decoupling economic growth from carbon dioxide emissions in China's manufacturing industry

·      Decoupling economic growth from carbon dioxide emissions in China's metal industrial sectors: A technological and efficiency perspective

·      Economic growth and carbon emissions: Estimation of a panel threshold model for the transition process in China

·      Can China meet its 2020 economic growth and carbon emissions reduction targets?

·      Determinants for decoupling economic growth from carbon dioxide emissions in China

The paper presents no methodology and methods. It is clear the paper is not defined on any scientific ground similar to the studies mentioned above.

To what extent is the paper quantitative, qualitative or mixed?

The extent to which the research questions tie to a defined methodology is not discussed or presented.

The paper should focus on primary data as there are too many studies that applied the adopted approach.

Thus, the paper is a repetition of similar approach as indicated above. 

The concept of using secondary data and running various statistical tools to create a story does not indicate gap. 

The findings of the study are consistent with similar papers as no new concept emerged, as it is a pattern of repetitive studies.

The paper presents no section to discuss the results.

Reviewer 2 Report

This paper focuses on a very interesting subject. That being said, there are some points that require attention. I would advise authors to consider the following comments:

  1. Which of the research findings relate to the journal’s aims and scope? Authors must make the necessary changes so that results are related to environmental stewardship, environmental medicine, and public health
  2. Criteria used for dividing the study area into 4 zones must be stated. Moreover, economic areas must be explained. Do these areas have a kind of administrative autonomy that enables decision-making?
  3. Discussion is integrated into results. Although this format is acceptable, it fails to discuss results adequately. In order to discuss results effectively, there should be some comparison with the theoretical background. At this stage, however, there is no effort to connect results with previous literature while no references in this part have been used.
  4. Study limitations should be added and recommendations for future studies must be made.
  5. Line 514 should be rephrased by paying more attention to explanations on policymakers. Who are they and which are their responsibilities?

Reviewer 3 Report

    This manuscript focused on the decoupling between economic growth and carbon emissions. The authors applied the Tapio decoupling index method and VAR model, divided China into four regions and obtained the impulse response and variance decomposition results. Finally, based on the results obtained, some policy comments were proposed. However, the following questions should be given more attentions and great response before the manuscript is accepted.

1. It is recommended that the manuscript be polished and checked in details, as there are some grammar errors in the present edition. Especially some long sentences should be carefully rewritten and organized. Such as line 12~16, 90~91, and 94~97.

2. The introduction section needs to be improved. The subsequent part does not respond to the issues raised in the previous section. The connections between sentences are loose and need to be improved to be more logical, rather than just putting the evidence needed into the text. As well as Sec. 2.2.

3. There are some points in the article that are not accurate, please check and correct them carefully. For example, in line 15, '20th century', and in line 232, 'each type is subdivided into eight decoupling states'.

4. The meaning of the abbreviation 'CI' in line 147 is not mentioned in the previous section.

5. As stated in the article, there are three methods to measure the decoupling status, why choose  the Tapio decoupling model?

6. The description of the results is not clear enough, for example, in lines 298 to 302, there is no description of the trend in renewable energy consumption and no indication of which parameter is making the difference smaller.

7. Line 146: 'Economic growth is the main factor for the increase of carbon emissions.' while the first sentence of the introduction says that large amounts of energy consumption achieve economic growth, and energy consumption is the main source of carbon emissions. Is it appropriate to attribute the increase in carbon emissions to economic growth in this sentence?

8. 'Energy Intensity' is mentioned several times in Sec. 2.1, but is not explained.

Round 2

Reviewer 1 Report

The paper has been improved to merit publication.

Author Response

Thanks for your comments.

Reviewer 2 Report

Dear authors,

The revised manuscript has been improved significantly and is written in an engaging and lively style. In addition, it provides a very useful resource for the examined topic. Hence, I have recommended the acceptance of the paper. However, I would advise the author to consider the following comment:

Incorporate the last paragraph of section 4.7 into section 5.3.

Author Response

Thanks for your comments. We have integrated the last paragraph of section 4.7 into section 5.3.

Reviewer 3 Report

The manuscript has been polised and suitable for the publication.

Author Response

Thanks for you comments.